# A Review of Recent Research into the Causes and Control of Noise during High-Speed Train Movement

Hongyu Yan [1,2,3], Suchao Xie [1,2,3,*] , Kunkun Jing [1,2,3] and Zhejun Feng [1,2,3]

1 Key Laboratory of Traffic Safety on Track, Ministry of Education, School of Traffic & Transportation Engineering, Central South University, Changsha 410075, China; 204201034@csu.edu.cn (H.Y.); jkk0711@csu.edu.cn (K.J.); 194201004@csu.edu.cn (Z.F.)
2 Joint International Research Laboratory of Key Technology for Rail Traffic Safety, Changsha 410075, China
3 National & Local Joint Engineering Research Center of Safety Technology for Rail Vehicle, Changsha 410075, China
* Correspondence: xsc0407@csu.edu.cn

**Abstract:** Since the invention of the train, the problem of train noise has been a constraint on the development of trains. With increases in train speed, the main noise from high-speed trains has changed from rolling noise to aerodynamic noise, and the noise level and noise frequency range have also changed significantly. This paper provides a comprehensive overview of recent advances in the development of high-speed train noise. Firstly, the train noise composition is summarized; next, the main research methods for train noise, which include real high-speed train noise tests, wind tunnel tests, and numerical simulations, are reviewed and discussed. We also discuss the current methods of noise reduction for trains and summarize the progress in current research and the limitations of train body panels and railroad sound barrier technology. Finally, the article introduces the development and potential future applications of acoustic metamaterials and proposes application scenarios of acoustic metamaterials for the specific needs of railroad sound barriers and train car bodies. This synopsis provides a useful platform for researchers and engineers to cope with problems of future high-speed rail noise in the future.

**Keywords:** high-speed train; noise control; acoustic metamaterial; sound barrier; rail vehicle

## 1. Introduction

The sound of trains is considered a disturbance to most residents around the railroad and train passengers, and the sound generated by trains can interfere with sleep, life, and work. As early as 1825, a letter from Leeds Intelligencer presents a record of train noise interfering with life [1].

Noise has always been a major threat to people's health, and several studies have shown that people exposed to noise for a long time have an increased risk of stroke [2], coronary heart disease [3], and many other cardiovascular and cerebrovascular diseases [4]. People who live in noisy environments for long periods also have significantly higher rates of endocrine disorders and breast cancer. Noise causes physical problems as well as sleep disturbances and mental problems [5].

Countries around the world have introduced various laws and regulations to protect people from noise hazards. The European Regional Environmental Noise Guidelines recommend that rail noise should be controlled to below 54 decibels (dB $L_{den}$). In 1974, the United States first enacted noise regulations to ensure the health of people. In China's railroad environmental noise management regulations, the railroad environmental noise emission standards implement Category 4b of the Sound Environmental Quality Standard (GB12525-90). This category requires no more than 70 decibels (dB $L_{eq}$) during the day and no more than 60 dB (dB $L_{eq}$) at night [6]. Japan promulgated noise standard values for the

Shinkansen in 1975. The values specified in decibels ($L_{pA}$.Smax) in these standards are 70 dB or less, mainly for residential areas, and 75 dB or less for other areas [7].

The rise in the industrial civilization of the nineteenth century brought the railroad, and people's pursuit of train speed drove the continuous development of train technology. Steam locomotives reached their peak in the 1930s when Gresley's streamlined "Mallard" locomotive reached a top speed of 202 km/h. In 1964, Japan's Shinkansen was the first to commercialize modern electric high-speed trains, operating at 210 km/h en route from Tokyo to Osaka.

Nowadays, the operating speed of high-speed trains has reached 350 km/h. In the future, the speed of high-speed trains will be further increased, and the noise problem will be further exacerbated, making the train noise problem a hot research topic among engineers with an interest in rail transportation. High-speed train noise is already a problem that cannot be ignored, and if this problem is not solved timeously, it will seriously affect the future development of high-speed trains.

To this end, this study presents a selective literature review focusing on:

(1)   The causes of high-speed train noise and the distribution of the sound sources;
(2)   The current main research methods for high-speed train noise;
(3)   Traditional methods of high-speed train noise control;
(4)   Potential uses of acoustic metamaterials in the area of high-speed train noise.

## 2. High-Speed Train Noise Composition

While the noise of early steam locomotives was generated by steam engines, the noise of modern electric high-speed trains consists of wheel-rail rolling noise and aerodynamic noise caused by train airflow [8].

### 2.1. High-Speed Train Aerodynamic Noise

The airborne noise generated by high-speed trains is divided into external airborne noise and internal airborne noise, of which the external airborne noise of high-speed trains is the main source of noise. When the driving speed of high-speed trains exceeds 300 km/h, the external airborne noise accounts for more than 50% of the total noise of the train [9], and as the speed of the train increases, the external aerodynamic noise of the train will increase in the ratio of six to eight times the speed of the train [10].

The aerodynamic noise sources are monopoles, dipoles, and quadrupoles. The aerodynamic noise generated by high-speed trains during the driving process is mainly caused by the dipole and quadrupole sound sources generated by the surface pressure fluctuations around high-speed trains. When running, the aerodynamic noise will change due to the speed and the surrounding environment [9].

The train's aerodynamic noise mainly comes from the pantograph area, bogie area, connection area, and the concave structural areas of the train's body surface. Smoothing the design of the train body [11], flow-field control, laying sound-absorbing material [12], and other measures are the main ways to reduce noise generation on the train body. Noise reduction in the pantograph region is currently achieved by optimizing the pantograph shape. Pantograph shape optimization can control the scale of eddy current shedding and thus reduce noise [13]. The addition of shields on both sides of the pantograph effectively diverts the airflow and prevents noise diffusion. The contribution of radiated noise from pantographs accounts for more than 10% of the total noise generated. Researchers mainly study the sound generation mechanism of pantographs and optimize them through experiments and numerical simulations. The bogie fairing can reduce the development of turbulence outside the fairing, thus reducing the noise from the bogie. Meanwhile, the use of a reasonable skirt design also reduces the influence of the train bottom spill on the car body and decreases noise propagation therefrom [14]. The inter-coach windshield region is considered one of the main sources of aerodynamic noise. At present, a fully enclosed outer windscreen is used to ensure that the external airflow and the inner windscreen do

not come into contact, thus eliminating the vortex in the inner windscreen part, which can control the aerodynamic noise generated on the windscreen of the train [15].

The pantograph is the main noise source of aerodynamic noise in high-speed trains. Compared with the cavity and bogie, the pantograph is located on the top of the train, and the aerodynamic noise generated by it is difficult to be isolated by sound barriers. This paper counts some representative pantograph noise reduction measures in recent years, as shown in Table 1, which are found through comparison and analysis. For the noise reduction measures in the pantograph area, the current research can be summarized into two levels: one is to reduce the number of rods of the pantograph for the pantograph itself; second, passive noise reduction is applied to the shunt area.

**Table 1.** Comparison of noise reduction measures for pantographs.

| Methods to Reduce Noise | Research Methods | Frequency Characteristics | Sound Pressure Level | Reference |
|---|---|---|---|---|
| Different strip spacing of the pantograph | CFD analysis | No effect | The sound pressure level of the standard noise measuring point is reduced by 2.8 dB with a spacing of 540 mm | [16] |
| Bionic pigeon feathers | CFD analysis | Around 1000 Hz (original model) Around 100 Hz (optimized model) | The total noise decreased by 10 dB | [17] |
| Pantograph insulators with elliptical section | CFD analysis | Tonal peaks are gradually reduced from 2 kHz | The peak sound pressure level of elliptical insulators decreased by 4 dB | [18] |
| Cylindrical rod and push rod applied to a layer of porous sound absorption material | CFD analysis, Wind tunnel test | No effect | The peak sound pressure level of the optimized pantograph decreased by 5 dB | [19] |
| Covering the fairings with a porous material | CFD analysis, Wind tunnel test | No effect | The optimized noise peak is 8 dB lower than the original noise peak | [20] |
| Covering the circular cylinder with metal foam | Wind tunnel test | Tonal peaks toward lower frequencies | The peak sound pressure level decreased by 5 dB at 216 km/h | [21] |
| Pantographs with and without the cavity | CFD analysis | No effect | The difference in OASPL between the pantographs with and without the cavity was approximately 4 dB at the side | [22] |
| The comparison of noise reduction effects for four types of pantographs covers | CFD analysis | No description | A pair of baffles with half of the height of the pantograph on both sides can lessen noise by about 3 dB | [23] |
| Noise contribution from high-speed train roof configuration of cavities, ramped cavities, flat roofs | CFD analysis | No effect | The flat roof with side insulation plates has the lowest overall noise levels. | [24] |

In Table 1, we analyze the noise reduction measurements, research methods, frequency variation, and sound pressure level variation of each research. Through comparison, we find that researchers generally study the pantograph noise reduction problem by wind tunnel tests and CFD analysis, among which CFD analysis is the most widely used research method, which indicates that CFD technology has strong application value in the field of high-speed trains at present. In some of the studies, the results of wind tunnel tests also fully prove the reliability of CFD analysis. In the future, the flow simulation methods and computer performance algorithms will be the focus area of the researcher.

At present, optimizing the rod structure of the pantograph is a feasible measure to reduce the pantograph noise. By analyzing the noise frequencies of each study, we can find

that optimizing the cylindrical rod and push rod of the pantograph will change the frequency of pantograph noise so that the peak frequency shifts to the low-frequency direction. Compared with high-frequency noise, low-frequency noise is more difficult to be absorbed by porous materials (polyurethane fiber, glass fiber wool, etc.). Using metamaterials to absorb low-frequency noise may be a feasible method. In Section 4, we further introduce the metamaterial for sound absorption. Bionic pigeon feathers shaped pantograph rods and covering the fairings with the porous material bring the most significant noise reduction effect, but the shape of the pigeon feathers will significantly increase the processing cost, and the porous material will change its pore structure under high wind conditions. Compared with other noise reduction measures, changing the strip spacing of the pantograph is the easiest to implement, but this has only been analyzed by CFD analysis, and the specific effect has yet to be experimented with.

An overall analysis of the studies in Table 1 shows that all pantograph noise reduction measures can improve the aerodynamic noise of high-speed trains to some extent, but the improvement is limited and difficult to be applied in actual high-speed train manufacturing. Perhaps there will be better measurements to improve the pantograph noise in the future, but at present, it is difficult to optimize the pantograph to reduce the noise. Therefore, in future research, improving the train body panels and sound barriers to stop noise propagation may be the best way to reduce aerodynamic noise.

### 2.2. High-Speed Train Mechanical Vibration Noise

### 2.2.1. Braking Noise

Brake squeal noise is loud and associated with high sound pressure and a sharp tone, which poses a hazard to people's physical and mental health. The friction of the brake disc generates brake squeal noise, friction generates wear, and the generation and development of brake noise are inseparable from frictional wear behavior.

In their study, Eriksson et al. [25] proposed that wear of the braking surface leads to significant randomness in braking noise spectra. Graf et al. [26] suggested that the uniform friction layer of the braking interface exerts an important influence on the occurrence of squeal noise. Renault et al. [27] estimated the effect between the surface morphological characteristics of the braking interface and the braking noise. Majcherczak et al. [28] discovered that material debris generated changes the friction coefficient and thus affects frictional noise. Massi et al. [29] found that when frictional squeal noise is generated during braking, a large amount of debris tends to accumulate on the material surface.

The aforementioned research found that high-speed train brake noise and tribology are inseparable, and now with the continuous improvement of materials and production processes of the train brake disc, frictional brake noise is further reduced.

### 2.2.2. Wheel-Rail Noise

Wheel-rail noise is mainly divided into three types: impact noise, rolling noise, and spike noise. When the train speed is less than 300 km/h, rolling noise is the main noise source [30]. Wheel-rail noise is generated by the relative motion between the wheels and the track due to the roughness of the rail [31]. The degree of wheel and rail roughness also determines the amplitude of vibration and other dynamic characteristics of each component during train travel. There is a mutual force between the wheels and the rails during motion of the train, and this force causes the rails and wheels to vibrate, thus radiating noise into the surrounding air. In 1976, to find the sound radiation between the wheel and rail, Remington et al. [32] proposed a simplified engineering method to predict wheel-rail noise by simplifying the wheel and rail as an infinitely long Eulerian beam and an Eulerian beam under continuous elastic support, respectively. Thompson considered the symmetry of the wheel based on Remington's work and calculated several typical wheels of that time by the finite element calculation method [33]. For the rail model, Thompson considered the influences of the rail sleeper [34] and wheel rotation [35], used the two-dimensional boundary element method to calculate the vibration sound radiation

efficiency, and developed the wheel-rail noise prediction software "TWINS" according to the research results.

With the continuous improvement of the wheel-rail noise prediction model, the technique to eliminate wheel-rail noise is becoming increasingly mature. Reducing wheel and track surface roughness, improving the wheel-rail design, and using local shielding can better solve the train wheel-rail noise problem. Jones et al. [36] added shields to the bogies; Vincent et al. [37] eliminated the noise from the train wheels by changing the train track and sleepers; Bouvet et al. [38] and Cigada et al. [39] proposed improvements to the wheels to suppress wheel-rail noise by increasing the elasticity of the wheels; and Merideno et al. [40] changed the modalities of the wheels using sandwich-type dampers, thus reducing vibration and noise. For train wheels, Lee et al. [41] improved wheel wear by optimizing the curvature of the wheel webs so that the wheels maintain a stable noise level after long-term use. In terms of train rails, Chen et al. [42] believed that replacing rails on viaduct lines with damped rails can reduce noise while changing the peak frequency band of the noise.

## 3. High-Speed Train Noise Research Methods

At present, the noise of high-speed trains in motion is mainly studied by line tests, wind tunnel tests, and numerical simulations.

### 3.1. Real High-Speed Train Noise Tests

Real high-speed train noise tests can obtain the most realistic noise distribution during the train-driving process. Real high-speed train noise tests are generally divided into internal and external noise acquisition. Internal noise experiments on high-speed trains generally use different locations in the car for microphone arrays for acoustic signal acquisition [43], but the internal noise data alone cannot be used to assess the overall noise level in service and its effect on the surrounding environment: it is now generally used in sound-insulation research on high-speed train bodies [44].

Train exterior noise tests generally involve measurement of the noise distribution in space through single or multiple acoustic sensors and can also obtain the far-field field noise information of moving high-speed trains. The total sound pressure of the high-speed trains during the driving process and the contribution of different frequency bands of noise to the overall high-speed train interior and exterior noise can be measured using the acoustic signals collected by microphones at different measurement points. The arrangement of the acoustic sensors affects the collected noise data. As shown in Figure 1, the one-dimensional horizontal arrangement of the acoustic sensor array identifies the distribution of sound sources in the direction of motion of the train. The one-dimensional vertical arrangement can identify the sound source distribution at the height of the train [45]. The X-shaped sensor array identifies noise data in both horizontal and vertical directions relative to the high-speed train, and these data can be used to map the sound source distribution of the high-speed train [46]. By increasing the number of acoustic sensors, the microphone array is also arranged in spiral, star, or spherical forms.

The number of microphones in the microphone array further increases with the increase in the speed of the train. He et al. [47] used a star-shaped microphone array with 78 microphones to test the sound field of a high-speed train in a real-world experiment. Li et al. [48] employed a microphone array consisting of 78 microphones to measure the noise levels of high-speed trains traveling over viaducts and embankment sections, and Zhang et al. [49] used a microphone combination consisting of a 78-channel microphone array and microphones at a horizontal height of 3.5 m to study the sound field of high-speed trains travelling at a speed of 350 km/h. Noh [50] used a microphone array with 144 channels to conduct sound field tests on a high-speed train running at 390 km/h. As shown in Figure 2, these train noise experiments collected real high-speed train noise data and used the data to study the noise distribution and variations therein.

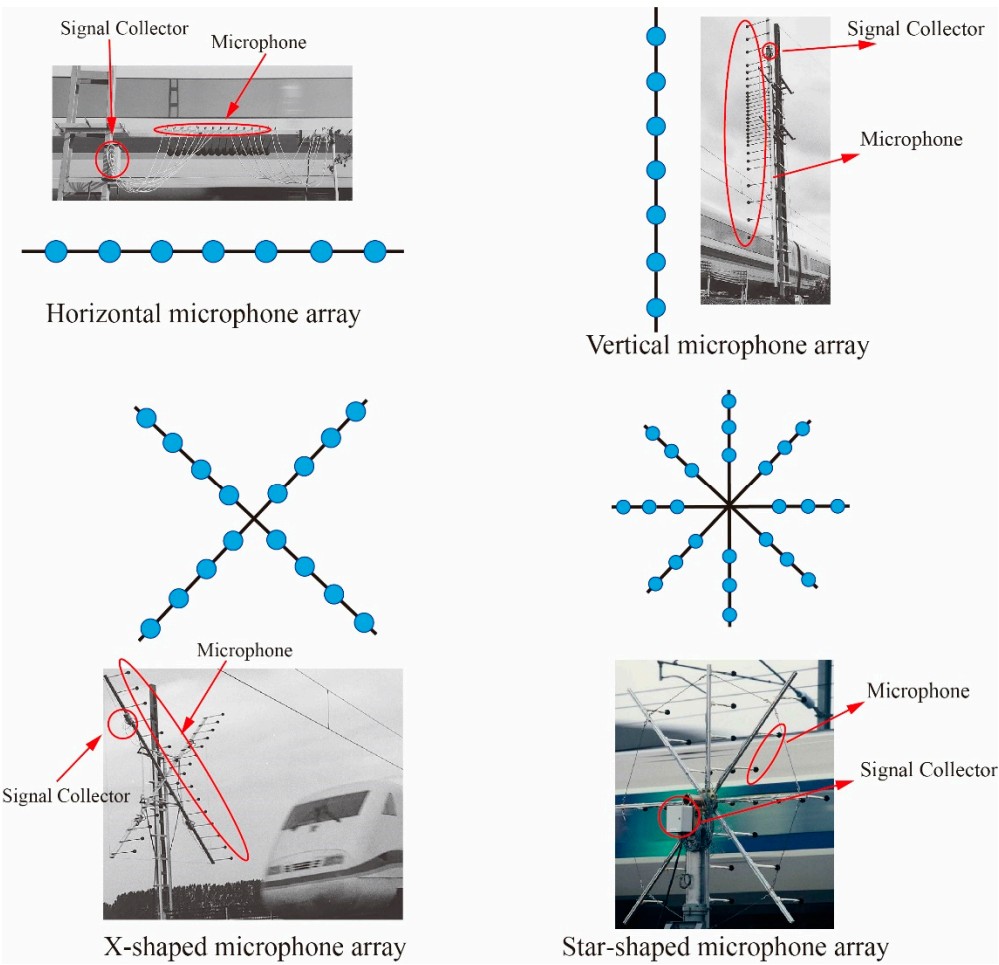

**Figure 1.** Microphone array arrangement form [45,46].

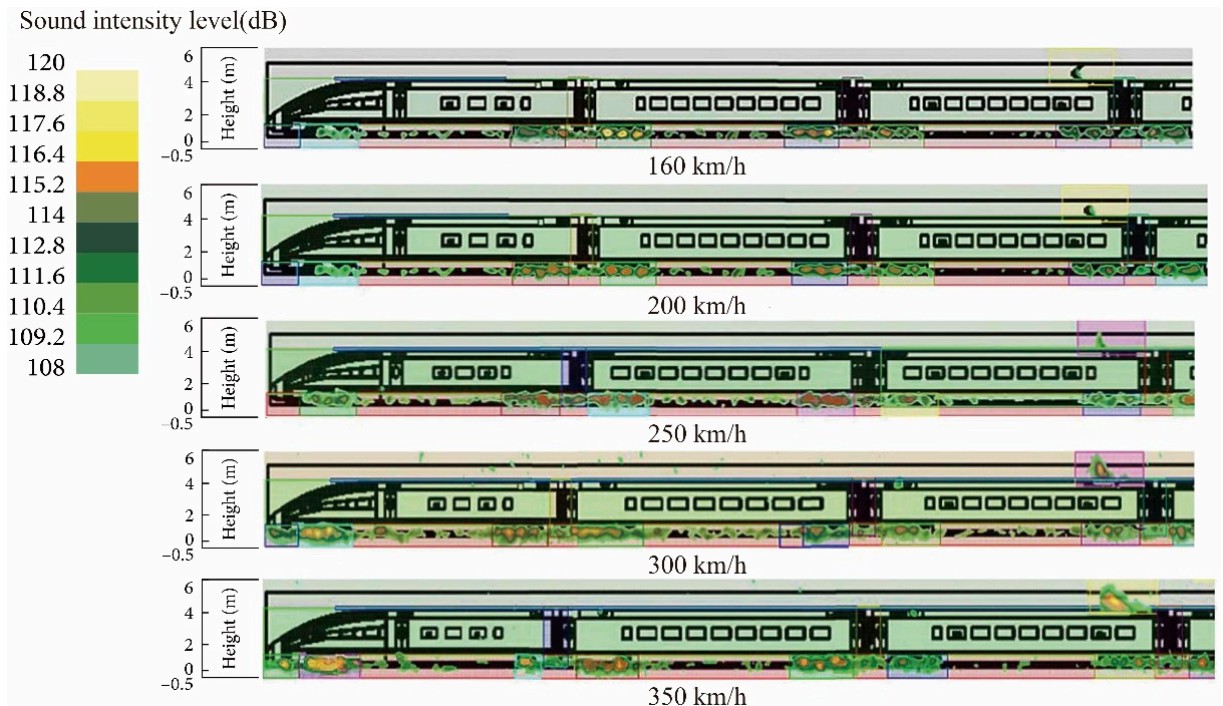

**Figure 2.** Noise distribution at different train speeds [48].

### 3.2. Wind-Tunnel Test

Compared to train noise experiments, wind tunnel experiments can be conducted in an indoor environment, independent of external climatic conditions, and can simulate external wind and other climatic conditions according to the hardware facilities of the wind tunnel, thus realizing the controllability of environmental factors. In aerodynamic research, wind tunnels are cutting-edge experimental testing tools. Both Ferrari and NASA have built wind tunnels to improve the aerodynamic performance of Formula 1 cars and space shuttles. As shown in Figure 3, wind tunnel facilities for high-speed trains typically employ large tunnels that use giant vanes at the entrance to generate strong airflow, with special grilles to reduce vortices in the airflow before it enters the laboratory. The noise collected in the wind tunnel test does not include mechanical noise, so it is possible to improve evaluations of the aerodynamic noise generated by the train.

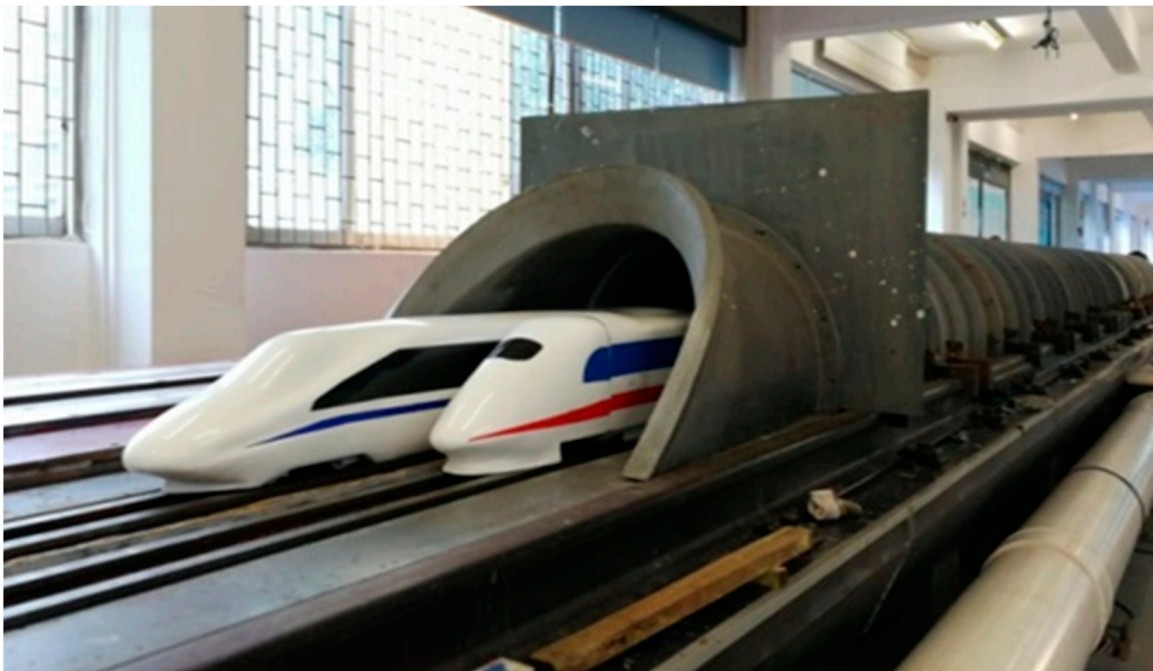

**Figure 3.** Train isometric model.

As shown in Figure 3, to reduce the cost of wind tunnel experiments, scaled-down models are often used for experiments. In 1991, Baker et al. [51] used a 1/76 scale HST train model and a 1/20 scale TGV001 train model to evaluate the effects of different types of ground simulations on train drag. Willemsen et al. [52] experimentally investigated three different types of trains at a 1/10 scale using a German-Dutch wind tunnel, and the results of the study showed that high Reynolds numbers on the surface significantly improve the reliability of wind tunnel experiments on trains. Nagakura et al. [53] conducted wind tunnel experiments using a 1/5 scale model of the Shinkansen train, and they estimated the contribution of each source of noise generated by a Shinkansen train to the roadside noise level based on the experimental data. Zhu et al. [54] proposed a numerical simulation method for wheel-track noise of the high-speed train and verified the accuracy of this method through wind tunnel experiments. Zhang et al. [55] performed a large eddy simulation to research the unsteady flow near the pantograph of the DSA380 high-speed train and predicted the aerodynamic noise caused by the pantograph. Although wind tunnel experiments are cheaper and have a higher safety factor, current wind tunnel experiments are more demanding in terms of experimental environment and equipment for non-full-size flow field simulations due to the similarity criterion.

### 3.3. Numerical Simulation and Theoretical Research

To conduct train noise tests or wind tunnel tests relative to the various complex environments through which high-speed trains pass is costly, but the full range of flow field information can be determined quickly and easily through computational fluid dynamics (CFD). Computational fluid dynamics has various advantages, such as fewer restrictions and lower costs. Using the method of computational fluid dynamics, it is possible to use mathematical methods to discretize the flow field control equations in the grid of the computational region, solve the discrete numerical solutions, and obtain the aerodynamic noise of the train during travel according to the Navier–Stokes equations. As shown in Figure 4, the high-speed train model is huge and has a complex structure, so the mesh must contain many elements, often too onerous given current computational resources, to satisfy the need for a direct solution for the whole vehicle model. In CFD commercial software, the most common method is to ascertain the computational flow field characteristics by using Lighthill's acoustic approximation model and then use the acoustic analogy theory to simulate the noise propagation, thus improving the computational efficiency and saving computational resources.

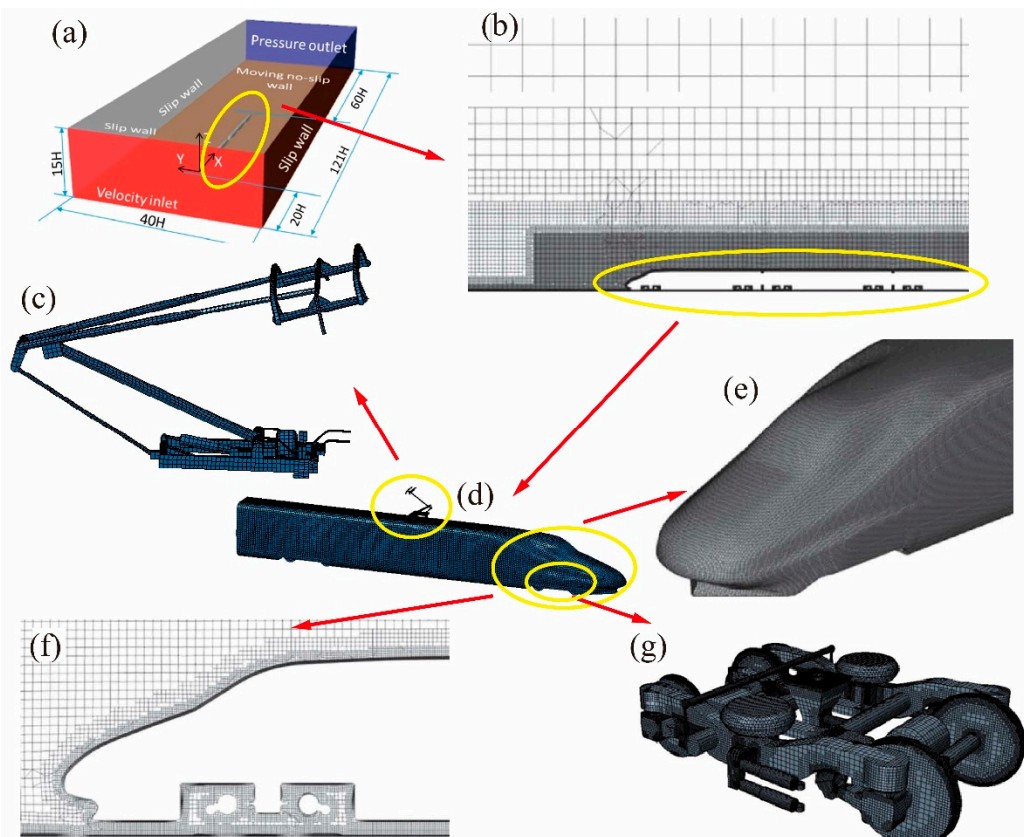

**Figure 4.** CFD meshing for high-speed trains. (**a**) Schematic of flow field computational domain [56]; (**b**) grid density variation in the computational domain [56]; (**c**) pantograph meshing [57]; (**d**) high-speed train complete grid [57]; (**e**) high-speed train head meshing [56]; (**f**) high-speed train head grid density variation [56]; (**g**) bogie meshing [58].

Many scholars have studied high-speed train noise using computational fluid dynamics. Sassa et al. [59] combined acoustic and fluid analyses to calculate the aerodynamic noise radiation from the surface vestibule side doors of high-speed trains. Takaishi et al. [60] derived the far-field integral equation for aerodynamic noise and calculated the aerodynamic noise conditions at the train pantograph. Masson et al. [61] developed a numerical model of the French TGV train based on the lattice Boltzmann method and obtained the noise distribution around a TGV high-speed train. Wang et al. [57] and Wu et al. [62] used large

eddy current simulation and boundary element method to simulate the pantographs of high-speed trains and found that the pantograph noise is concentrated in the low-frequency range, and the pantograph aerodynamic noise is gradually dispersed in the high-frequency range. Sun et al. [63] conducted a simulation-based analysis of the pantograph and found that the slide plate, pantograph head, balance rod, insulators, bottom frame, and pull rod are the main sources of aerodynamic noise thereon. By comparing six models, Li et al. [64] found that the SST k-ω method is the most suitable for numerical simulation of train aerodynamic behavior in crosswinds. To cope with the problem of insufficient computational resources during the numerical computation of high-speed trains, Liu et al. [65] proposed a model for pantograph noise prediction based on the Reynolds number provided by a single component.

As shown in Figure 5, CFD software can give you the calculation results of running train vortex shedding. The rapid development of computers in the future will further promote the development of numerical simulation and theoretical research into high-speed trains to provide more abundant solutions to the noise problems associated with high-speed trains.

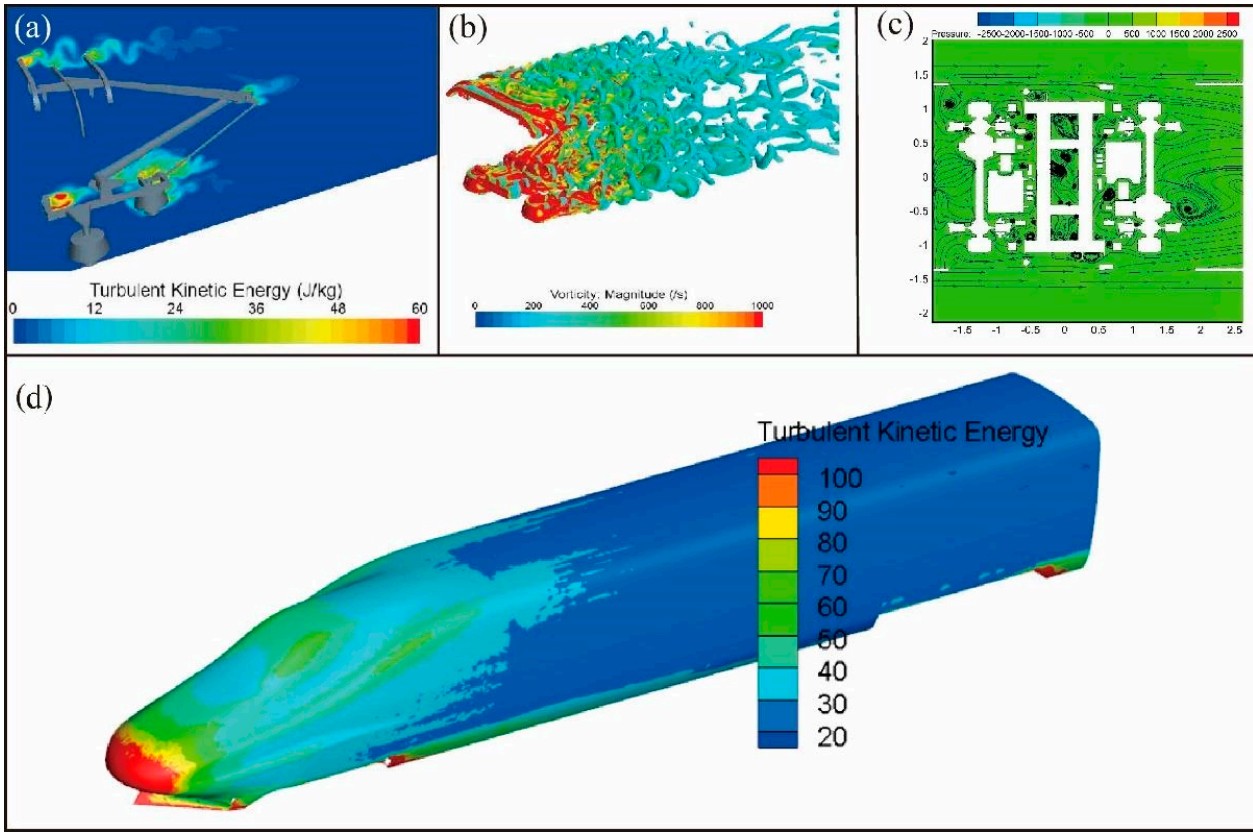

**Figure 5.** (**a**) Turbulent kinetic energy distribution of the pantograph [55]; (**b**) instantaneous iso-surface plots of the Q-criterion, colored by vorticity magnitude (Q = 10,000) [55]; (**c**) bogie speed flow chart [58]; (**d**) turbulent kinetic energy distribution of high-speed trains.

## 4. High-Speed Train Noise Control Methods

### 4.1. Traditional Noise Control Methods for High-Speed Trains

#### 4.1.1. Train Vibration Reduction

High-speed trains will inevitably impose a greater impulse on railroad infrastructure, causing an increase in the vibration intensity of the infrastructure and the environment along the railroad line [66]. Under the action of the cyclic force caused by high-speed trains, excessive vibration amplitudes can damage the infrastructure, while this vibration also generates much noise, affecting the safety, comfort, and stability of train operation [67].

At present, the main active isolation measures and passive isolation are adopted to reduce high-speed train vibration and thus control high-speed train noise. Active isolation measures include floating slabs [68], highly resilient rail pads [69], and high-performance wheels [70]. All these measures can directly reduce the vibration of the train to a large extent. Passive isolation measures include open trenches and soft-filled barriers [71]. Although the above measures show a good damping effect, damping to reduce the actual engineering problems related to noise still faces certain difficulties due to the huge mass and fast speed of high-speed trains.

### 4.1.2. Train Body Sound Insulation

The noise inside the high-speed train is mainly transmitted from outside. The sound source and vibration excitation outside the train can be transmitted to the inside of the train through two main paths: airborne sound and structural sound transmission, thus forming the noise inside the train [72]. Therefore, improving the sound insulation performance of the high-speed train carriage panel structure is conducive to improving the acoustic environment inside the car. The train carriage panels are mainly extruded aluminum profile structures (floor, roof, and sidewalls) and double-plate cavity structures (windows). At present, extruded aluminum profiles are the main sheet structure of high-speed train bodies, and their acoustic performance is one of the important factors influencing the acoustic environment inside the vehicle.

Due to the special structural form of extruded aluminum profiles, their sound insulation performance is poor. Improving the sound insulation performance of aluminum profiles is a key focus of much research in academe. Xin et al. [73] studied the sound insulation characteristics of orthogonally rib-stiffened sandwich structures and corrugated core sandwich structures and the mechanism of sound radiation during the vibration of these structures. Xie et al. [74] developed an acoustic model of the statistical energy method for the amount of sound insulation of extruded profiles, and the model was able to favorably predict vibration excitation. Kim et al. [75] improved the sound insulation performance of aluminum profiles by filling. Qin et al. [76] employed an optimized finite element energy statistics method (FE-SEA) and verified its efficacy. Li et al. [77] used a waveguide finite element (2.5-d FE) model combined with the energy statistics method to investigate the sound insulation model of extruded aluminum profiles. Zhang et al. estimated the coupling loss factors between the structure and cavity and established a statistical energy analysis model to make predictions of bogie noise and bottom plate sound transmission. Based on the model, they developed a program that can quickly design the internal panel structure of a train and proposed a model that can predict the sound absorption and insulation characteristics of the composite bottom panel of a train [78,79]. Thereafter, they proposed to improve the noise insulation of the train by adding high-damping rubber and water-based damping as a coating to the train floor [80,81].

For the double-plate cavity structure of the window part of the train, Xin et al. [82] systematically studied the sound insulation characteristics of the finite and infinite double-plate cavity structure using the wave propagation method. Zhang et al. [83] established and validated their sound transmission loss model of the window by measuring the vibration and the interior noise level of a window of a high-speed train.

### 4.1.3. Railroad Sound Barriers

A sound barrier featuring easy installation and obvious noise reduction is an important method of traffic-noise management, which has been widely used and developed throughout the world [84]. Sound barriers mainly include wide top types, semi-enclosed types, fully enclosed types, etc. The diversity of sound barriers makes it possible to have different classification methods, which are based on appearance, line form, material, acoustic performance, structural form, etc. According to the differences in line form, the sound barriers can be classified into bridge sound barriers and road sound barriers; according to the differences in unit plate material, the sound barriers are classified into metal and

non-metal sound barriers. As shown in Figure 6, the common structures of sound barriers are upright insert types, T types, inverted L types, Y types, multiple edge types, etc. [85]. In practical application, insert-type sound barriers are most widely used. May et al. [86] used proportional model experiments to compare the noise reduction effects of upright, T-shaped, and Y-shaped sound barriers and proposed, for the first time, that T-shaped sound barriers provide the best noise reduction effects. Defrance et al. [87] simulated the noise reduction effect of T-shaped sound barriers with sound-absorbing materials using the boundary element method and undertook experimental verification thereof. Baulac et al. [88] improved the acoustic performance by providing slots at the top of T-shaped sound barriers and optimizing the form and disposition of the slots using genetic algorithms. Oldham et al. [89] investigated the factors affecting the noise reduction effect of T-shaped sound barriers using numerical calculations and found that the additional noise reduction of the top structure is related to the locations of the source and receptor of the sound and the location and height of the sound barrier. Venckus et al. [90] studied the roof-inclination angle of upright-type sound barriers and found that the sound barrier was most effective in reducing high-frequency sound waves when the roof inclination angle was 120°. Zhang et al. [91] proposed a semi-enclosed sound barrier with slits and verified its sound insulation effect. At present, sound barriers provide good control of environmental noise along the railroad, but the noise frequency band they control is singular, and because most of them are reflective sound barriers, they will aggravate the noise inside the train instead.

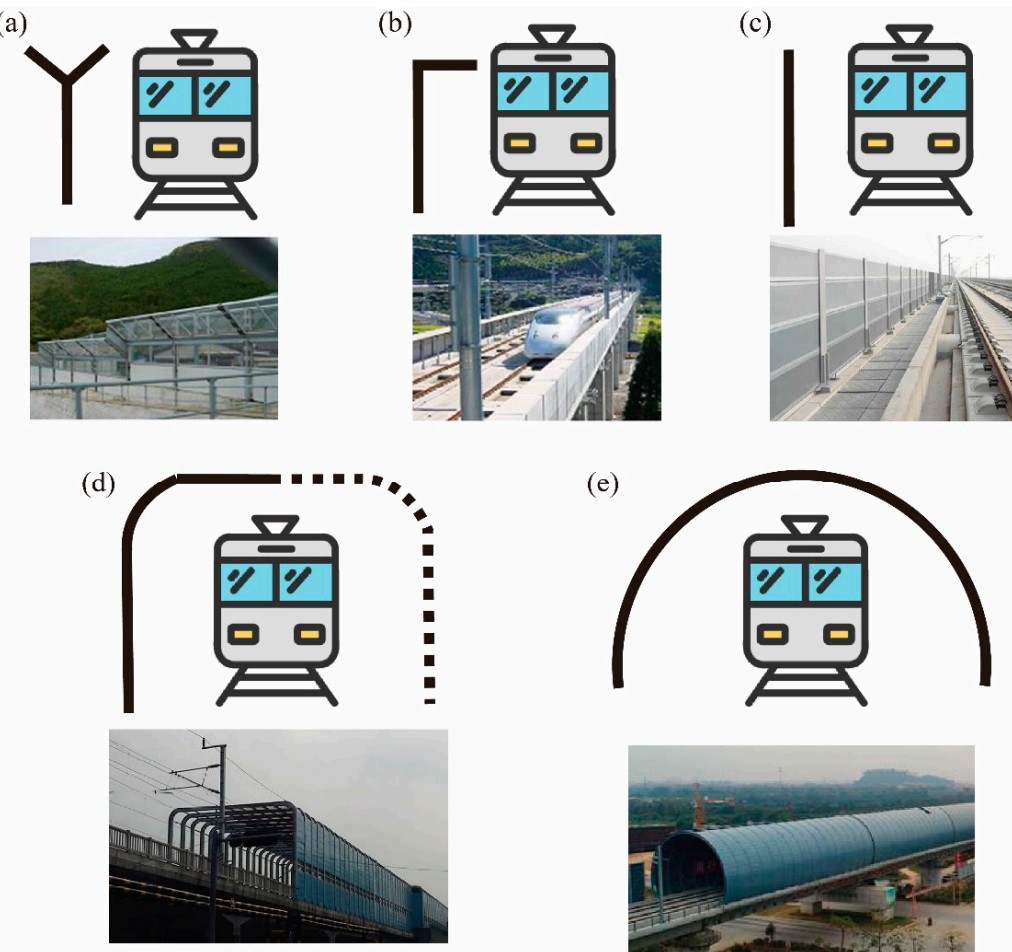

**Figure 6.** (**a**) Y-shaped sound barrier; (**b**) inverted L-shaped sound barrier; (**c**) upright type sound barriers; (**d**) semi-enclosed sound barrier; (**e**) fully enclosed sound barrier.

### 4.2. Acoustic Metamaterials Applied to High-Speed Train Noise Control

After the train speed is increased to 600 km/h, the aerodynamic noise and wheel-rail noise will be significantly increased, and the noise-reduction ability of the traditional extruded aluminum profiles used in the body of high-speed trains, the sound barriers along the railroad line, and the polyurethane foam materials laid inside the car body are all stretched to their operational limit. When traditional materials and structures fail to meet the needs of high-speed train development, acoustic metamaterials with exotic characteristics will become an important means by which to solve the noise problems associated with high-speed trains.

#### 4.2.1. Definition and Development History of Acoustic Metamaterials

The term "metamaterial" is commonly used to describe artificial composites consisting of periodic or randomly arranged artificial subwavelength structures, a concept first introduced by Veselago in 1968 in the field of electromagnetism [92]. The emergence of electromagnetic metamaterials sharpens researchers' understanding of metamaterial theory, and the concept of metamaterials was also introduced in the fields of optics, mechanics, and heat transfer. The concept of acoustic metamaterials can be traced back to Narayanamurti [93], who first discovered that periodic structures could be used to control high-frequency phonon propagation but did not refer to such periodic structured materials as acoustic metamaterials. In 2000, Liu et al. [94] designed a small ball made of high-density lead wrapped in rubber and then proposed the theory of locally resonant phonon crystals using this model, opening the door to the study of acoustic metamaterials. After this, acoustic metamaterials have been further developed by artificially designing microstructures to allow materials to present limitations beyond their original natural laws. This can realize a series of idiosyncratic material functions, including acoustic stealth [95], acoustic directional transmission [96], acoustic negative refraction [97], acoustic focusing [98], low-frequency sound absorption [99], and so on. At present, acoustic metamaterials have been applied to solve engineering problems such as aircraft cabin noise reduction, automotive NVH, and building facades and have achieved better vibration and noise suppression effects [100].

Metamaterials often have different properties from those of traditional materials. For natural materials, material parameters such as mass density, Young's modulus, and Poisson's ratio are positive in the natural case, while for artificial structures, effective material parameters may become negative within a specific frequency range. Some mechanical metamaterials have been applied in the field of energy-absorbing structures and body structures of high-speed trains [101,102], and the application of metamaterials to reduce railroad noise will be one of the important means of dealing with the noise problem associated with future high-speed trains.

#### 4.2.2. Metamaterials Applied to Railroad Sound Barriers

At present, railroad sound barriers are mainly upright insert-type sound barriers. The wind pressure fluctuations, when passing through such sound barriers, can lead to the loosening and breaking of bolts and the destruction of sound barrier panels [103]. When designing high-speed railroad sound barriers, not only should the sound insulation and sound absorption characteristics of the barrier be considered, but also the dynamic response of the sound barrier structure. Conventional sound barriers produce sound reflections that lead to increased noise levels within high-speed trains. For future high-speed railroad sound barriers, they inevitably need to meet codified sound insulation criteria and, at the same time, have both ventilation and noise absorption capabilities.

There is always a balance between the thickness of a sound insulation device and its ventilation capacity. Zhang et al. [104] designed a binary structure consisting of a coiled unit and a hollow tube with a thickness of less than one-fifth of the wavelength, which can block low-frequency sound from different directions while allowing 63% of the airflow to pass through. Wu et al. [105] designed a vented metamaterial absorber operating at low frequencies (<500 Hz) with only two absorption units, achieving high-efficiency absorption

under vented conditions by using weak coupling of two identical split-tube resonators. Huang et al. [106] combined spiral channels and embedded apertures; this metamaterial structure can absorb low-frequency noise while maintaining the requisite thickness. Ghaffarivardavagh et al. [107] proposed a deep sub-acoustic wavelength metamaterial cell that includes nearly 60% of the open area allowing passage of air and also serves as a high-performance selective sound muffler. Wang et al. [108] proposed an acoustic metamaterial composed of many cells, and this open metamaterial contains a large hole in each cell to ensure airflow. Kumar et al. [109] integrated eight labyrinth cells of different configurations to form an acoustic metamaterial and introduced a herringbone channel to achieve ventilation. Xie et al. [110] proposed a metamaterial with a conchoidal cavity structure and embedded this metamaterial into a conventional concrete or metal sound barrier, which can significantly improve the sound absorption capacity of the sound barrier and prevent sound pollution of the surrounding high-rise buildings by high-speed railroads.

As shown in Figure 7, acoustic metamaterials can ensure better sound insulation and absorption effects under ventilation, but they have not been widely used in sound barriers due to their complicated production process and high cost of production. To increase the potential for wider engineering applications of acoustic metamaterials, researchers should further simplify their structure while considering the production cost in the design process and, furthermore, fit the specific use scenario for optimization.

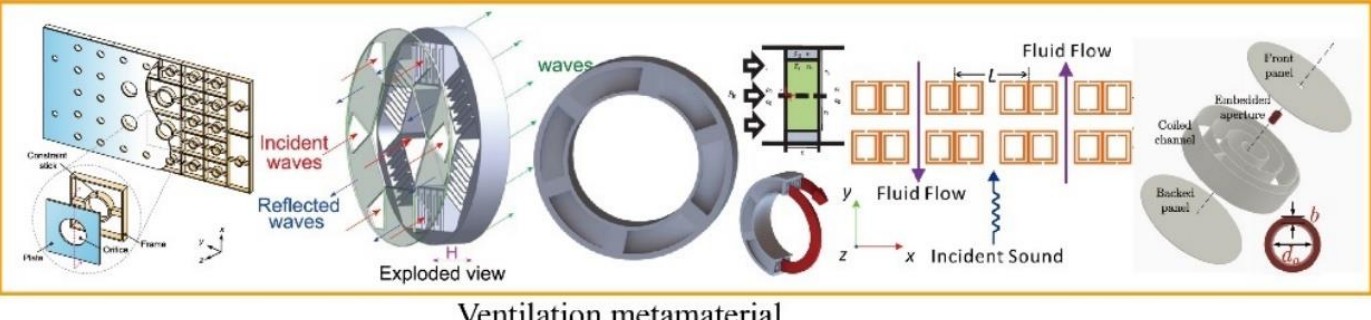

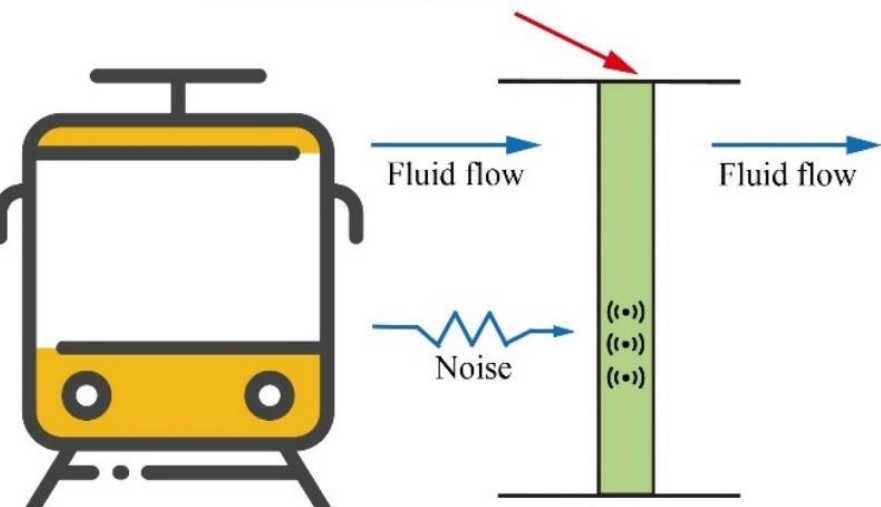

**Figure 7.** Acoustic metamaterials with ventilation properties applied in sound barrier.

### 4.2.3. Application of Metamaterials in Train Car Bodies

At present, the car body of high-speed trains is mainly fitted with a mixture of damping pulp and fibrous materials as sound insulation and vibration damping materials. At present, the train body shows poor sound insulation and absorption ability, especially for the long wavelengths of low-frequency noise, which passes more easily through the body structure and body gaps into the train interior. This impairs the quality of the acoustics within the train interior and adversely affects passenger comfort.

As shown in Figure 8, scholars have explored the application of acoustic metamaterials in engineering. Li et al. [111] combined microperforated plates and embedded partitions to design a controllable broadband acoustic unit with a thickness of only 70 mm. Liu et al. [112] added a separator plate with micro-perforations inside the cavity of a common Helmholtz resonator to demonstrate the absorber a multi-order absorption capability and expanded the absorption bandwidth of the absorber by combining eight absorbers. Long et al. [113] designed a multi-band quasi-perfect absorber constructed by a double-channel Mie resonator, which is more flexible, to achieve multi-band quasi-perfect absorption. The combination of labyrinth structure and micro-perforated plate structure often can produce better sound absorption. The labyrinth structure can partition the cavities, while the combination of different cavities and perforated plates can yield acoustic metamaterials with a wider frequency range. Zhang et al. [114] designed an acoustic metamaterial consisting of a single-hole plate and a labyrinth cavity combination, which achieved good sound absorption in the low-frequency range. Liu et al. [115] investigated an acoustic metamaterial that achieved nearly perfect sound absorption in the range of 380 to 3600 Hz by combining a variety of different micro-hole plates and labyrinth cavities. The honeycomb structure is the best topology covering two-dimensional planes with good mechanical properties and is often used as the core structure of high-speed train underlayment. Tang et al. [116] modified the traditional honeycomb sandwich panel by introducing micro-pores based on the honeycomb-corrugated hybrid core to acquire an acoustic metamaterial with good sound absorption in the low-frequency range. Peng et al. [117] designed a composite honeycomb structure by combining different microporous and honeycomb cavities to fabricate an acoustic metamaterial with 90% sound absorption in the range of 600 to 1000 Hz. Wang et al. [118] proposed a NOMEX honeycomb metamaterial with acoustic absorption capability based on NOMEX honeycomb, which can achieve quasi-perfect absorption against noise in high-speed train motion. Wu et al. [119] designed a hybrid metamaterial absorber based on a microperforated plate and a coiled Fabry channel, which can achieve more than 99% sound absorption at the resonant frequency (<500 Hz) of acoustic absorption. Xu et al. [120] designed a metamaterial consisting of three holes and three cavities connected in parallel and investigated the effects of different temperatures on its acoustic absorption. Xie et al. [121] added polyurethane material to the acoustic metamaterial composed of microporous plates and cavities to achieve continuous ultra-broadband acoustic performance.

Figure 9 analyzes the main noise sources of high-speed trains, and it can be found that the wheel-rail area noise, pantograph noise, and inter-coach gaps noise all rise to a great extent as the speed of high-speed trains increases. Among the three main noise sources, the pantograph noise increases most significantly with speed, and when the speed reaches 386 km/h, its main noise level exceeds that of the wheel-rail area and inter-coach gaps. The frequency variation of the main noise sources also deserves attention; compared with the noise in the range of 2000 Hz–4000 Hz, the improvement is more obvious in the range of 500 Hz–2000 Hz. The pantograph noise changes more significantly with speed, and the noise level in the 500 Hz–1500 Hz range reaches nearly 100 dB(A) after the speed reaches 386 km/h. The significant increase in noise levels in the mid-frequency and low-frequency can seriously affect passenger comfort.

Traditional acoustic materials mainly use porous structures, such as acoustic sponges, felt, glass fiber, polyurethane foam, etc. When sound waves pass through the pores of various materials, their kinetic energy is converted into thermal energy, leading to the dissipation of sound wave energy to attenuate noise; however, porous materials have poor absorption capacity for low-frequency noise, and the material thickness is strictly limited due to the size of the train, which makes it difficult for the current acoustic insulation car body design to deal with low-frequency noise. In contrast, metamaterials, with their unique material properties, can absorb low-frequency noise while maintaining a small thickness.

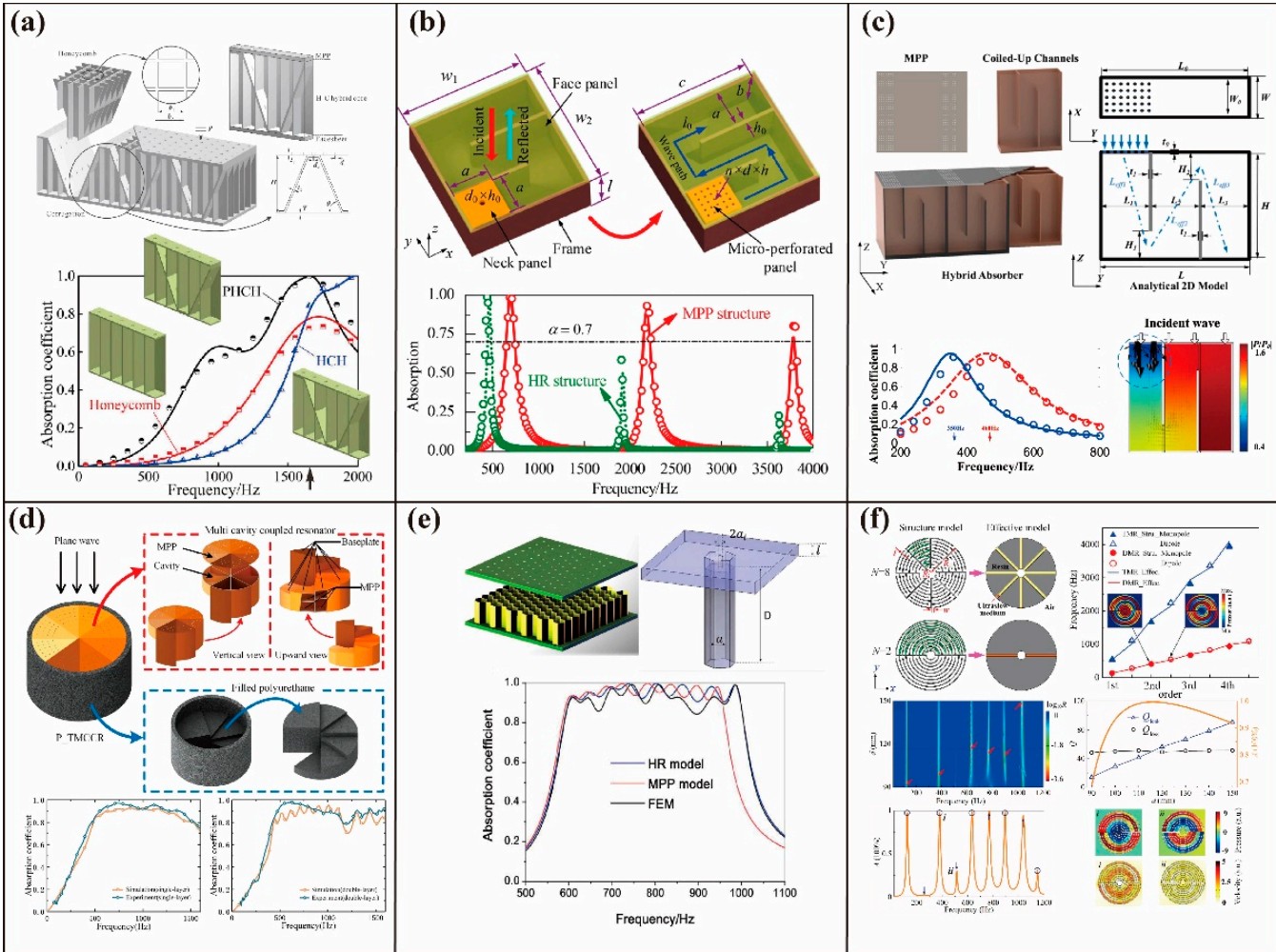

**Figure 8.** (**a**) Schematic of perforated honeycomb-corrugation hybrid (PHCH) metamaterial [116]. (**b**) Schematic of the broadband metamaterial unit [115]. (**c**) Schematic diagram of a hybrid absorber with microperforated plates and coiled channels [119]. (**d**) Schematic representation of a tunable multi-cavity coupled-resonator with polyurethane-filled slits [121]. (**e**) Schematics of the composite honeycomb sandwich panels [117]. (**f**) Multiband quasi-perfect low-frequency sound absorber based on double-channel Mie resonator [113].

According to the latest research on acoustic metamaterials shown in Figure 8, it can be found that acoustic metamaterials can cope with low-frequency and mid-frequency noise better and ensure a high noise absorption level in a thinner case. Figure 8d,e of the absorption coefficient curve can be found that in acoustic metamaterials in a wide range of frequencies, the sound absorption ability can reach more than 80%, and this acoustic characteristic can just meet the noise problems faced by high-speed trains. At present, acoustic metamaterials have been widely used in the fields of aerospace, ships, automobiles, and buildings. It is believed that acoustic metamaterials, with their excellent acoustic properties, will also become an important measure to solve the noise problem of high-speed trains.

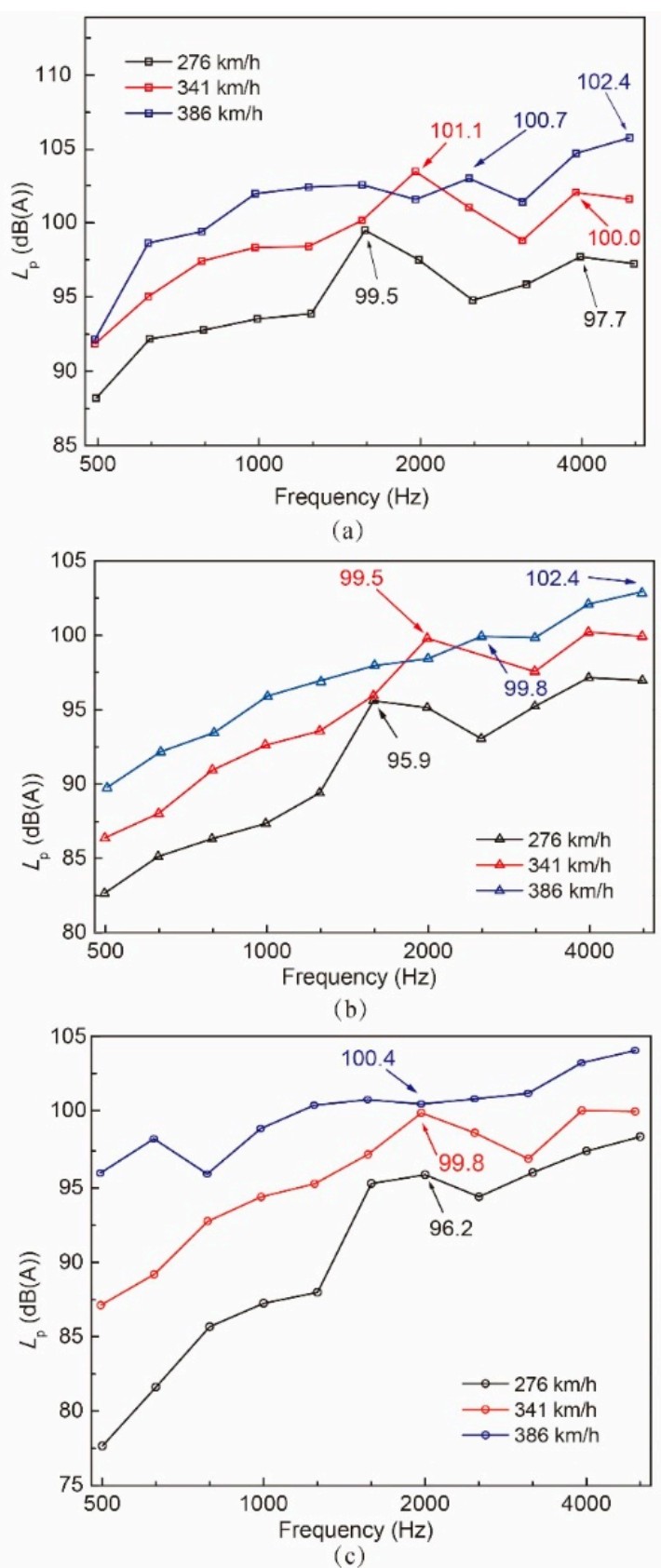

**Figure 9.** The main noise frequencies characteristics of high-speed trains: (**a**) wheel-rail area; (**b**) inter-coach spacing; (**c**) pantograph [47].

### 5. Summary and Prospect

With the increase in train speed and the development of modern train structures, the noise problem has become more pressing, and solutions thereto more complicated. The development of high-speed trains also brings new opportunities and challenges for future train noise research. To cope with the train noise problem, it is necessary to follow the development of science and adopt new materials, new research tools, and new theories. In this review, we have identified the following problems and made corresponding suggestions:

(1) The main component of high-speed train noise has gradually changed from mechanical vibration noise to aerodynamic noise, and with the further increase in running speeds in the future, the aerodynamic noise will be further intensified. The structure of the high-speed train pantograph, high-speed train skirt, and high-speed train connection should be further optimized, and the noise problem should be emphasized in the optimization process.

(2) The acoustic insulation level of traditional railroad sound barriers is limited and vulnerable to air pulsation stress. The introduction of acoustic metamaterials can largely improve their acoustic insulation performance, but at present, the structure of acoustic metamaterials is complicated, and their processing cost is high, so research on their manufacturing process and batch production is warranted.

(3) It is difficult for the current body structure of high-speed trains to cope with low-frequency noise, and the narrow body structure cannot be arranged with thicker materials to realize sound absorption and insulation. Thus, future acoustic metamaterials may be the best means of achieving sound absorption and insulation in the train body.

**Author Contributions:** Conceptualization, H.Y. and S.X.; methodology, H.Y.; software, H.Y.; formal analysis, H.Y., K.J. and Z.F.; writing—original draft preparation, H.Y., S.X., K.J. and Z.F.; writing—review and editing, H.Y., S.X., K.J. and Z.F.; supervision, S.X.; funding acquisition, S.X. All authors have read and agreed to the published version of the manuscript.

**Funding:** This research was funded by the National Natural Science Foundation of China (51775558). This paper was also supported by the Nature Science Foundation for Excellent Youth Scholars of Hunan Province (Grant No. 2019JJ30034) and the Shenghua Yu-ying Talents Program of the Central South University (Principle Investigator: Suchao Xie).

**Institutional Review Board Statement:** Not applicable.

**Informed Consent Statement:** Not applicable.

**Data Availability Statement:** Not applicable.

**Acknowledgments:** This research was undertaken at the Key Laboratory of Traffic Safety on Track (Central South University), Ministry of Education, China. The authors gratefully acknowledge the support from the National Natural Science Foundation of China (Grant No. 51775558). This paper was also supported by the Nature Science Foundation for Excellent Youth Scholars of Hunan Province (Grant No. 2019JJ30034) and the Shenghua Yu-ying Talents Program of the Central South University (Principle Investigator: Suchao Xie).

**Conflicts of Interest:** The authors declare no conflict of interest.

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
