# Peer review of "A Review of Recent Research into the Causes and Control of Noise during High-Speed Train Movement"

_applsci, doi:10.3390/app12157508_

Round 1

Reviewer 1 Report

The paper was a pleasant read with an impressive amount of references (117!).

The content of the paper provides a rather high-level overview, that can be considered as a short “textbook”. This may be expected because the title indicates that a review is intended. However, for a scientific journal paper it may lack some depth. Additionally, both external and internal noise aspects are considered, which are complicated topics for such a short paper.

Perhaps the authors can include a section to elaborate more a specific recent noise control measure? This can be the external or internal noise aspect.

Some minor comments/questions:

-        All 4 authors are from the same 3 (!) organizations?

-        In the Introduction 54 and 75 dB are mentioned. Are these the same quantities? (e.g. Lden, Leq or Lmax). Also, it would be nice to mention the noise standard values for the Shinkansen, for at least one category.

-        The 4 topics in the Introduction are mapped to Sections: 1 -> Sec. 2, 2 -> Sec.3, 3&4 -> Sec.4

-        The use of “air noise”. Use “flow noise” or “airborne noise”?

-        “wheel and rail unevenness”. Use “roughness”?

-        Section 2.2.2 provides a part on noise control. Move to Section 4?

-        Section 3. Mention and list (ISO) standards on railway noise?

-        In general, provide a reference in the text to the figures. Also, are some figures too small?

-        Section 4.1.1. What is meant by “active isolation”? (actuators and a power supply is used?)

-        Section 4.1.2. How to use “water-based damping as a coating to the train floor”?

-        Section 5: Summary and prospect’s’? (or outlook)

-        Section 5. For a barrier the acoustic “insulation” is mentioned. Is also the noise (reduction) over the barrier meant? Increasing the insulation can be done easily by adding weight, such as the use of concrete.

-        Section 5. What are “super-materials”? (first mentioned here.)

Author Response

The revised manuscript and response comments have been combined in the attachment.

Reviewer 2 Report

The paper proposed is well organised. It gives a large overview of the studies carried out on environmental noise emitted by high speed trains. It is a complete bibliography but there is no technical/scientific analysis. There is n added value of the paper. For example, it can be interesting to put together all the data collected and to analyse them. Then it will be possible to extract the main characteristics of the noise sources (position on the train, mean level, spectrum...) depending of the train type.

The paper must be improved by adding some analysis of collected information to provide an added value.

Author Response

We have revised the manuscript, and the revised manuscript and response comments are in the attachment.

Reviewer 3 Report

The paper as written and presented does not look like a scientific work, but it looks like a compilation work (the papers of other authors are listed).

Authors should explain their contribution to the evolution of the subject.

The problem of train noise is a very extensive and complex subject.
